

# Measurement Report: Source apportionment and environmental impacts of VOCs in Lhasa, a highland city in China

Chunxiang Ye[1], Shuzheng Guo[2], Weili Lin[2], Fangjie Tian[3], Jianshu Wang[1], Chong Zhang[1], Suzhen Chi[1], Yi Chen[2],
Yingjie Zhang[1], Limin Zeng[1], Xin Li[1], Duo Bu[4], Jiacheng Zhou[5], Weixiong Zhao[5]

[1] State Key Joint Laboratory for Environmental Simulation and Pollution Control, and College of Environmental Sciences and Engineering, Peking University, Beijing, 100871, China.
[2] Key Laboratory of Ecology and Environment in Minority Areas (Minzu University of China), National Ethnic Affairs Commission, Beijing 100081, China
[3] Senior Department of Cardiology, the Sixth Medical Center of PLA General Hospital, Beijing, China
[4] Science Faculty, Tibet University, Lhasa 850000, China
[5] Laboratory of Atmospheric Physico-Chemistry, Chinese Academy of Sciences Hefei Institutes of Physical Science Anhui Institute of Optics and Fine Mechanics, Chinese Academy of Sciences, Hefei, 230031, Anhui, China.

*Correspondence to*: Chunxiang Ye (c.ye@pku.edu.cn)

**Abstract.** Hypoxia and adverse health outcomes might be complexed with $O_3$ pollution in the highland city of Lhasa. $NO_x$ emissions can amplify the role of volatile organic compounds (VOCs) in the secondary production of $O_3$ under the conditions of high ultraviolet (UV) radiation levels and unfavourable dispersion patterns in the Lhasa River valley. Here, online C2-C11 VOC measurements, accompanied by other parameters concerning the $O_3$ chemical budget, were first obtained and employed to identify the key VOC species and key sources of VOCs in terms of the loss rate against OH radicals ($L_{OH}$), ozone formation potential (OFP), secondary organic aerosol potential (SOAP) and toxicity. Oxygenated VOCs (OVOCs) were not only the most abundant VOCs but also dominated $L_{OH}$, OFP and toxicity. Isoprene and anthropogenic VOCs were further identified as precursors of these OVOCs. Aromatics accounted for 5% of the total VOCs (TVOCs) but contributed 88% to the SOAP and 10% to the toxicity. As the primary oxidative intermediates of aromatics were not well characterized by our measurements, the environmental impact of aromatics could be underestimated by our data. Source appointment and ternary analysis of benzene, toluene, and ethylbenzene confirmed the combined contribution of traffic emissions, solvent usage and biomass burning. This suggests that $O_3$ precursors are mainly from residents' life associated sources, except for solvent usage emissions which contribute to aromatic. Preliminary comparisons between source spectrums of transport sector emissions with PMF decomposed ones and our measured ones suggest that vehicle emission patterns of VOCs at



high altitude generally follows the ground-level impression. More quantitative data is required to further confirm this point though. Emission

reduction strategy analysis for $O_3$ pollution control highlighted multiple benefits of the simultaneous reduction in $NO_x$ originating from diesel vehicle emissions and biomass burning and background sources (possibly dominated by incense burning). The notable biogenic emission contribution to the OFP was also first confirmed in our study, and this highlights the side effects of the government's pursuit of a greener city.

**Summary**

Enhanced anthropogenic NOx and VOC emissions in the urban area of highland city of Lhasa promote local ozone photochemical generation

potential and increase the risk of photochemical pollution. Online C2-C11 VOC measurements by GCMS, accompanied by other parameters concerning the $O_3$ chemical budget, were first employed to identify the key VOC species and key sources.

The concentrations of the TVOCs ($18.70 \pm 8.35$ ppb) and major anthropogenic alkanes and aromatics measured in this study are approximately half of those measured in the megacity of Beijing, but several folds or even more than one order of magnitude higher than those measured at most regional measurement sites across the TP, confirming the anthropogenic contribution to VOCs. OVOCs were the most abundant (52% of the

TVOCs) and acetaldehyde was even comparable to the megacity. Alkenes (mainly isoprene) and OVOCs accounted for over 90% of the $L_{OH}$ and over 80% of the OFP, suggesting their key role in perturbing the photochemistry of $O_3$. Aromatics further contributed to 13% to the OFP and dominated SOAP. PMF decomposed six factors namely diesel vehicle emissions, solvent usage, plant and second generation, NG/LPG, gasoline vehicle emissions and biomass burning and background.

Source appointment and ternary analysis of benzene, toluene, and ethylbenzene confirmed their combined contribution by traffic emissions,

solvent usage and biomass burning. Transport sector emissions and biomass burning and background emissions should be targeted for their contributions to VOCs, NOx and BC. Solvent usage emission control is meaningful for the reduction of aromatic concentration.



dominated SOAP. PMF decomposed six factors namely diesel vehicle emissions, solvent usage, plant and second generation, NG/LPG, gasoline vehicle emissions and biomass burning and background. Transport sector emissions and biomass burning and background emissions should be targeted for their contributions to VOCs, NOx and BC.

NOx and VOC emissions in the highland city of Lhasa promote ozone photochemical generation potential. Online VOC measurements by GCMS, accompanied by other O3 precursors, were employed to identify the key VOC and key sources. The concentrations of the TVOCs ($18.70 \pm 8.35$
ppb) and major anthropogenic alkanes and aromatics measured in this study are approximately half of those in the megacity of Beijing, but nearly one order of magnitude higher than those at most regional measurement sites across the TP. OVOCs consist of 52% of the TVOCs. Alkenes (mainly isoprene) and OVOCs account for over 90% of the LOH and over 80% of the OFP. Aromatics further contributed to 13% to the OFP and dominated SOAP. PMF decomposed six factors namely diesel vehicle emissions, solvent usage, plant and second generation, NG/LPG, gasoline vehicle emissions and biomass burning and background.

## 1 Introduction

Over 3 million residents live in highland cities across the Tibetan Plateau (the TP) with an average altitude over 4 km. Hypoxia and adverse health outcomes (Bigham and Lee, 2014) pose a threat to local residents and tourists. Moreover, economic development has inevitably led to the emission of anthropogenic pollutants, and the air quality has consequently deteriorated in the absence of carefully designed and effectively complemented air pollution control strategies. Air pollution, complexed with $O_2$ deficiency, might interfere with human high-altitude adaptation
and cause more serious health issues for the human body. Proper attention should be given to the emission sources and their contributions to air pollution. However, very sparse measurement data are available for high-altitude regions, with fewer research efforts and complex required logistics as causes. A review summarized pollutant measurements at background mountain sites and concluded that there occur very weak natural and anthropogenic perturbations in local photochemistry but underlined the long-range transport of long-lived species (Okamoto and Tanimoto, 2016). However, in highland cities, notable anthropogenic perturbations in photochemistry and air pollution occur. Among these studies in
highland cities, the distribution of aerosol and long-lived source tracers and their linkage with industrial and constructive activities have been extensively examined, whereas the sources, photochemical fate and roles of reactive species, such as $NO_x$ and volatile organic compounds (VOCs), in $O_3$ pollution accumulation have been less notably considered.

Pollutant measurements at background TP sites have suggested low abundances of reactive species, which is a key feature of the background atmosphere. A mixing ratio of $NO_x$ ranging from tens to low hundreds of pptv has been recorded thus far only at the Waliguan (WLG) Global
Baseline Watch Station along the edge of the northeaster boundary of the TP (Ma et al., 2013; Xue et al., 2013). At the same pristine station on



the TP, the most abundant VOCs are long-lived VOCs, of which their concentrations are comparable to those at other alpine background sites worldwide, while reactive VOCs are less abundant than those at other alpine background sites (Xue et al., 2013). These measurements of both $NO_x$ and short-lived VOCs across the TP strengthened the impression of weak perturbations on local photochemistry by natural and anthropogenic emissions. Later, the characterization of regional VOCs across the TP clearly demonstrated the increasing influence of long-range

transport along the edge of the TP boundary (Li et al., 2017; Zhao et al., 2020). The promotion of the $O_3$ photochemistry by transported VOCs has not been carefully explored. However, the net $O_3$ photochemical production established at even background sites on the TP might be associated with transported VOC sources (Xue et al., 2013; Zhao et al., 2020).

Lhasa is the largest city on the TP and has been progressively growing in size and population over the last two decades. Surrounded by mountains with elevations over 5500 m above sea level (a.s.l.), the city resides in the Lhasa River Basin at 3600 m a.s.l., trapping and accumulating locally

emitted pollutants within the surrounding atmosphere. The city is not green or hot, and high biogenic emissions are therefore not notably considered. Pollution involving concentrations of fine particulate matter ($PM_{2.5}$), $O_3$ and their primary precursors, such as $NO_x$ and VOCs, originating from anthropogenic sources (such as vehicles, religious activities, and cooking and coal combustion processes) have drawn research attention. Primary emissions of aerosols stemming from biomass burning, fossil fuel combustion, and suspension dust are the major sources of $PM_{2.5}$ pollution in Lhasa, ever since the first aerosol data set became available for the urban district two decades ago (Huang et al., 2010; Liu et

al., 2013; Li et al., 2019; Zhao et al., 2022). Old-day aerosol sources, such as the burning of wood, agriculture residuals and cow dung cake and waste incineration, have been largely suppressed since the Action Plan for the Prevention and Control of Air Pollution issued in 2013. Dust components are another major fraction of $PM_{2.5}$, but have not experienced similar reduction as biomass burning. The secondary chemical composition of aerosols includes a small mass fraction of $PM_{2.5}$ in this highland city, in contrast to other areas of China (Cong et al., 2011; Li et al., 2019). Consequently, the annual mean mass concentration of $PM_{2.5}$ of 20 µg m$^{-3}$ has experienced a slow declining trend (Yin et al., 2019),

although extremely high $PM_{2.5}$ concentrations have not been observed ever since 2013. Overall, pollution measurements in Lhasa have recorded atmospheric change from relatively low anthropogenic emissions to intensive unregulated anthropogenic emissions and to finally emission reduction over the last 2-3 decades (Yu et al., 2001; Ran et al., 2012; Guo et al., 2022; Yu et al., 2022a). Compared to the levels three decades ago, enhanced photochemical production of $O_3$ and therefore aggravated $O_3$ pollution were recorded in 2012 (Ran et al., 2014). Aggravated $O_3$ pollution even lasts to the present day, with a slower increasing slope over the last few years (Yin et al., 2019). While comprehensive reactive

VOC measurements were not available until recently, the increased abundance of $NO_x$ from the sub-ppbv level in 1998 to a level higher than 10 ppbv in 2012 was identified as the major cause of promoted $O_3$ production. The sporadic availability of $O_3$ and $NO_x$ observations from 1998 to 2015 has impeded our understanding of the historic trend of $O_3$ and $NO_x$ and therefore the $O_3$ chemical regime. In contrast to the past, recent changes in the emission inventory and pollutant abundance are more traceable. Since the implementation of effective pollution control policies in



2013, such as eliminating yellow-label vehicles and old vehicles and providing liquefied petroleum gas (LPG)-fuelled central heating alternatives
to replace small coal-fired boilers, the emission inventory of $O_3$ precursors might dramatically shift from past levels. For example, LPG-fuelled combustion processes might promote the high-temperature generation of $NO_x$ while reducing the emission of primary aerosols and VOCs originating from biomass burning. Moreover, the government has organized continuous monitoring of various air pollutants including $SO_2$, $NO_2$, CO, $O_3$, $PM_{2.5}$, and particulate matter with an aerodynamic diameter of 10 microns or smaller ($PM_{10}$) since 2015, also providing a potential opportunity to better understand the current trend of $O_3$ and its precursors and their relationships. VOC measurements in Lhasa, especially since
2014, have been reviewed to compile a comprehensive data set for detailed $O_3$ photochemistry analysis. Three measurements of this kind have been documented thus far, to our knowledge. Two-hour resolution canister sampling and gas chromatography–mass spectrometry (GC–MS) analysis of VOCs from 9:00 to 19:00 facilitated $O_3$ chemical regime analysis with an emphasis on anthropogenic VOC emission reduction (Yu et al., 2022b). Silica cartridge measurements of oxygenated VOC (OVOC) photochemical tracers, such as formaldehyde, acetone, glyoxal and methylglyoxal, confirmed the large OH reactivity contribution of VOCs and the production of OVOC intermediates in the urban atmosphere of
Lhasa (Li et al., 2022). The only online VOC measurement study employed the headspace technique with GC and detection via a photoionization detector and flame ionization detector (GC–PID/FID) to accurately measure C2-C12 alkanes, alkenes, and aromatics at the city centre (Guo et al., 2022). Further source apportionment using the positive matrix factorization (PMF) model indicated that the traffic sector and background biomass burning are two major sources of the measured VOCs. However, OVOC intermediates were not measured via offline GC–MS or online GC–PID/FID methods.

Under the umbrella of the second Tibetan Plateau Scientific Expedition and Research Program (STEP), a series of field campaigns, referred to as @Tibet campaigns, were carried out from 2019 to 2022. The major objectives of @Tibet campaigns were characterizing the atmospheric chemistry over the TP and exploring its impacts on climate and air quality. In the summer of 2021, a comprehensive field campaign was conducted at Lhasa, refer to herein as @Tibet 2021, to understand the atmospheric oxidative capacity, $O_3$ pollution, and secondary aerosol formation in the urban environments in the TP. During @Tibet 2021, a comprehensive dataset including $NO_x$, CO, $O_3$, individual $NO_y$ species,
VOCs, BC, and chemical composition of $PM_{2.5}$ was collected. Herein, VOC tracers together with other source tracers including $NO_x$, CO, and BC were intensively examined to identify key VOCs, major sources of VOCs, and their diel profiles in terms of the loss rate against OH radicals ($L_{OH}$), ozone formation potential (OFP), secondary organic aerosol potential (SOAP) and toxicity.



## 2. Experiment

### 2.1 Measurement site

Continuous online observations were conducted from May 22 to June 10, 2021, in the yard of the Lhasa Branch of the Institute of Qinghai-Tibetan Plateau Research, Chinese Academy of Sciences (29.63 ˚N, 91.02 ˚E, 3640 m a.s.l.). The observational site is located west of urban Lhasa (Fig. S1), approximately 8.0 km east of the famous Potala Palace. To the north of the observational site is Jinzhu Road, a major road in Lhasa, which runs from east to west and is separated from the site by buildings and green spaces. Farms and green spaces are located to the south of the observational site. During the observation period, the dominant wind direction was from the west.

### 2.2 Measurement of VOCs and other pollutants

VOCs in ambient air were measured online with a temporal resolution of 1 h using a GC–FID/MS analytical system (Shimadzu GCMS2010, Japan) coupled with low-temperature preconcentration equipment (Peng Yu Chang Ya, Beijing, China). Among the VOCs, C2-C5 compounds (13 species) were separated in a PLOT-Al$_2$O$_3$ chromatographic column (15 m × 0.32 mm inner diameter (ID) × 3 μm; J&W Scientific, USA) and then analysed via FID measurement; the other compounds (85 species) were separated using a DB-624 column (60 m × 0.25 mm ID × 1.4 μm; J&W Scientific, USA) and then analysed via single-quadrupole MS detection (EI source, 70 eV) in the selected ion monitoring (SIM) mode. One complete cycle of the analytical process includes preparation, sampling and preconcentration, injection and analysis, and back-blowing, and its detailed principle is described in Wang et al. (2014). During the preparation period, the air sample was continuously pumped through the sampling pipe to maintain fresh air. Before VOC enrichment, the sampling air was passed through a H$_2$O trap at -60 ˚C and CO$_2$ trap and O$_3$ trap separately. After 30 min of preconcentration at a sampling flow rate of 10 mL/min, the trap was heated to 110 ˚C within 3 min, and the released VOC samples were injected into the columns for 1 min. The ramp-up procedure for the column temperature included initial maintenance at 35 ˚C for 3 min, followed by heating at a rate of 6 ˚C min$^{-1}$ to 180 ˚C and maintenance at 180 ˚C for 2 min. After analysis completion, N$_2$ was purged in reverse into the trap at a flow rate of 100 mL/min for 1.6 min, and the preparation period for the next cycle was then started.

The C2-C5 compounds detected via FID were quantified through the external standard method, and the other components detected via MS were quantified through the internal standard method. Four compounds, namely, bromochloromethane, 1,4-difluorobenzene, chlorobenzene-d5 and bromofluorobenzene, were used as internal standards. Multipoint calibrations were performed involving a mixture of 57 Photochemical Assessment Monitoring Station (PAMS) compounds (Linde Co., Germany), Environmental Protection Agency (EPA) TO-15 OVOCs and halogenated hydrocarbons (Spectra Gases Inc., USA) at 5 concentrations. The R$^2$ values of the calibration curve for the VOCs ranged from 0.995 to 1.000, indicating that the integrated areas of the peaks are proportional to the corresponding concentrations of the target compounds. A mixing gas with a 2 ppbv concentration was used for daily span assessment.



BC was measured using an AE33 Aethalometer (Magee Scientific Corporation, Berkeley, CA, USA) with a temporal resolution of 1 min. Surface NO/NO$_x$, O$_3$ and CO were measured with TE42itl, TE49i, and TE48itl instruments (ThermoFisher Scientific, Waltham, MA, USA), respectively. Meteorological parameters, including the temperature and relative humidity (RH), were measured by a meteorological sensor (HMP155A, Vaisala, Finland), and the wind speed and wind direction were recorded by a sensor (010C-1/020C-1, Metone, USA).

**2.2 Canister sampling of VOCs and measurement**

Suma canisters (Entech Instruments Inc., USA) were also used for VOC sampling and measurement. Prior to sampling, cleaning and pretreatment of the canisters were performed in accordance with EPA Method TO-15 using a cleaning device (Entech Instruments Inc., USA) and humidified nitrogen. After that, all evacuated containers were refilled with pure nitrogen, stored in the laboratory for at least 24 hours, and then analysed using the same methods as the field samples to evaluate contamination in the containers. Teflon membrane filters (with a pore size of 4.5 μm)

were used before the inlet to protect the steel canisters from dust and airborne particles. Negative-pressure sampling of whole-air samples was performed at Lhasa in different locations, including tunnels, diesel vehicle emission-impacted roadsides, refuelling stations, renovation sites, and incense burning locations. The samples were analysed via GC–MS within 10 days.

**2.3 L$_{OH}$, OFP, SOAP and toxicity assessment**

L$_{OH}$ for each component of the VOCs can be calculated as follows:

$\quad$ L$_{OHi}$ = $K_{OHi}$×[VOC]$_i$ $\qquad\qquad\qquad\qquad\qquad\qquad\qquad$ (1)

where $K_{OH}$ is the rate constant for the reaction of VOCs with OH (Atkinson and Arey, 2003)

The OFP of each VOC component was calculated using the maximum incremental reactivity (MIR) method to assess the contribution of VOCs to O$_3$ generation. The OFP can be calculated as follows:

$\quad$ OFP$_i$ = MIR$_i$×[VOC]$_i$ $\qquad\qquad\qquad\qquad\qquad\qquad\qquad$ (2)

where MIR is the maximum incremental reactivity constant obtained from the California Code of Regulations 2010 (Carter, 2010), and [VOC]$_i$ is the concentration of each VOC component in ppb.

The SOAP for the generation of secondary organic aerosols from each component of VOCs can be calculated as follows (Derwent et al., 2010):

$$\text{SOAP} = \frac{Increment\ in\ SOA\ mass\ concentration\ with\ species\ i}{Increment\ in\ SOA\ with\ Toluene} \times 100 \qquad\qquad (3)$$

This method selects toluene as the basic compound for the SOAP, which is an index expressed relative to toluene = 100, and the SOAP for other

compounds is expressed relative to toluene.

The relative toxicity effect and integrated effect assessment can be expressed as follows (Niu et al., 2016):

$\quad$ Relative Toxicity Effect = [VOCs]×Toxicity Grade $\qquad\qquad\qquad\qquad$ (4)

$\quad$ Intergraded Effect = 0.4×OFP+0.4×SOAP+0.2×Toxicity $\qquad\qquad\qquad$ (5)





### 2.4 Source apportionment via PMF

The PMF model 5.0 of the US EPA with a multivariate linear engine ME-2 platform was used to perform VOC source apportionment, similar to Guo et al. (2022).

In PMF, the input files consist of two matrices, the observed concentration of $x_{ij}$ and the uncertainty $u_{ij}$ (i denotes the observed samples and j denotes the species). The uncertainty in the data can be calculated according to Equation 6, where the data with concentrations higher and below the method detection limit (MDL) involve different calculations. Based on Equation 7, PMF calculates the following parameters: (1) the source

number, p; (2) the chemical composition of each source factor, f; (3) the contribution of each source factor to the sample, g; and (4) the residual $e_{ij}$ of each species to each sample. In Equation 7, g and f should be higher than 0.

$$u = \begin{cases} \sqrt{(Error\ Fraction \times Conc.)^2 + (0.5 \times MDL_j)^2} & Conc. \geq MDL \\ \frac{5}{6} \times MDL & Conc. < MDL \end{cases} \qquad (6)$$

$$x_{ij} = \sum_{k=1}^{p} g_{ik} f_{kj} + e_{ij} \qquad (7)$$

The goal of PMF is to minimize the function residual Q (Equation 8), where n and m are the number of samples and the number of species,

respectively.

$$Q = \sum_{i=1}^{n} \sum_{j=1}^{m} \left[ \frac{\sum_{k=1}^{p} g_{ik} f_{kj} + e_{ij} - \sum_{k=1}^{p} g_{ik} f_{kj}}{u_{ij}} \right] \qquad (8)$$

After the files were entered, species with S/N ratios lower than 0.2 or with more than 80% of the data below the MDL were defined as bad data and excluded from the model. Species with an S/N ratio ranging from 0.2 to 2 or with more than 20% of the data below the MDL were defined as weak data. Three to ten factors were estimated to obtain the best resolution based on statistical indicators, such as the Q value, change rate of

$Q_{true}/Q_{exp}$, residual distribution, and coefficient of determination. Possible source explanations were also considered to obtain reasonable results. The Fpeak value was set to range from -5 to 5, and the lowest change in Q was returned at Fpeak = 0. No visible correlations were produced between factors through the inspection of G-space plots.

## 3. Results and Discussion

### 3.1 General description

The surface wind field follows the orientation of the Lhasa River valley along the east–west direction. West winds dominated over east winds during our measurements, affecting our measurement site upwind of the Lhasa city centre for most of the measurement time (Figs. 1 & S1). The wind speed mostly did not exceed 4 m s$^{-1}$. No precipitation except for that on June 8 was recorded. Horizontal transport and vertical mixing, but not abrupt wet deposition, were therefore responsible for the removal of local pollutants. The topography of the Lhasa River valley also promotes the accumulation of secondary pollutants in the surrounding atmosphere.




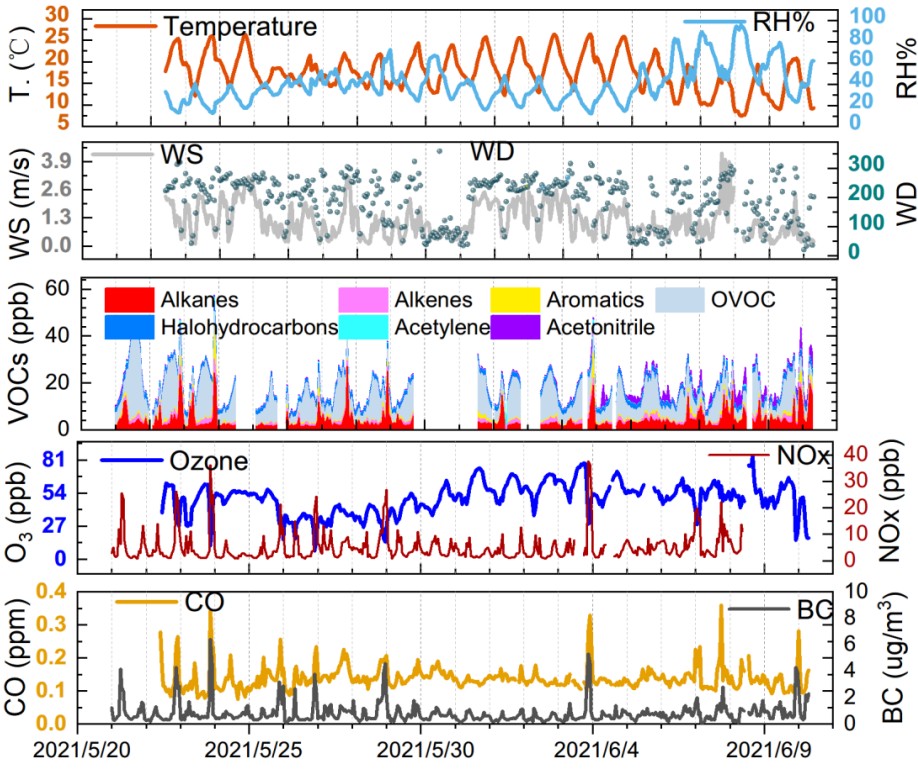

**Figure 1: Time-series variations in hourly averaged meteorological parameters, VOCs and inorganic tracers during the three-weeks measurements in**
**Lhasa. Dash line marked the mid-night of the day.**

$O_3$ varied between 15.6 and 84.0 ppb with an hourly mean value of 49.6±12.9(±1sd) ppb and a median value of 51.5 ppb. A typical noontime

peak at approximately 58.5 ppb was observed, suggesting that notable photochemical production of ozone occurred. $O_3$ precursors are not only

abundant but also reactive in this highland city. The $NO_x$ concentration varied between 0.73 and 37.58 ppb, with a noontime $NO_2$ minimum of


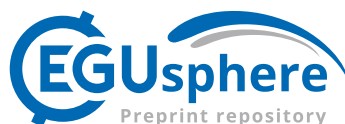

0.45 ppb and a noontime $NO_2$ photolysis lifetime of 2-11 minutes. Early morning and later afternoon peaks of $NO_x$ indicated the major contribution of traffic emissions. Both the CO of $137.7\pm35.5$ ppb and BC levels of $0.8\pm0.6$ µg m$^{-3}$ were much higher than background levels (Zheng et al., 2017; Zhao et al., 2020), confirming the influence of the transport sector and biomass burning. The TVOC concentration reached $18.70 \pm 8.35$ ppb. The online GC–MS VOC measurements highlighted that a considerable fraction of the total VOCs (TVOCs) comprised OVOCs (Figs. 1 & 2a). Specifically, OVOCs accounted for the largest proportion (52%) of the TVOCs (Fig. 2b), while alkanes, alkenes,

aromatics, and halohydrocarbons accounted for 22%, 6%, 5% and 11% respectively. The major OVOC, except for MTBE, are listed in Table 1. Most OVOCs were the photochemical intermediates of alkanes and less abundant alkenes, such as isoprene (Atkinson and Arey, 2003; Mellouki et al., 2015). The much higher abundance of OVOCs than that of their precursors suggests that photochemical decay of alkenes, long-chain alkanes in the highland environment lead to effective photochemical accumulation of these measured OVOCs. Not only concentration of acetaldehyde is comparable to that in Beijing summer, but also the (MVK+MACR)/isoprene ratio of 1.5 are compared with or higher than those

values in Beijing summer (Table 1). A high abundance of reactive OVOCs could also promote the photochemical production of $O_3$ and SOA, in addition to these primary VOCs. In our case, only small OVOC molecules (C ≤ 6) were recorded via online GC–MS; hence, $O_3$ production, rather than SOA formation, could be better induced. Even though, the considerable abundance of aromatics and expected photochemical intermediates of aromatics (not fully measured here) also indicated a high potential for SOA production.

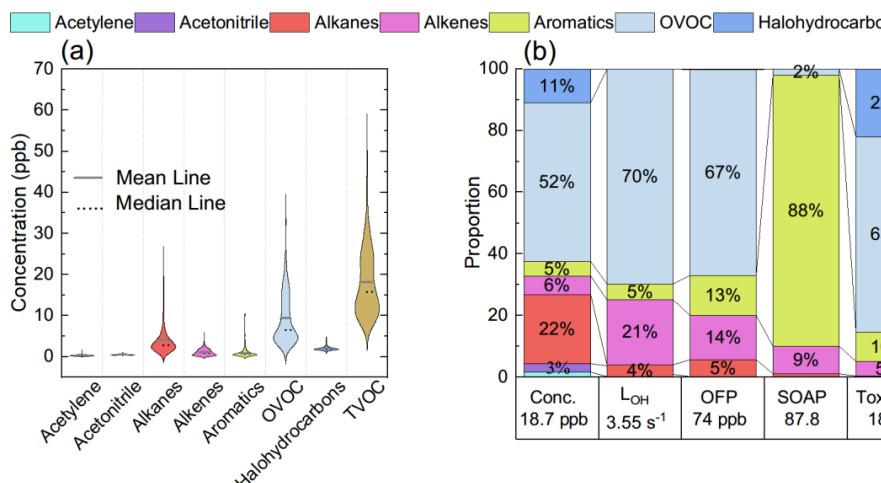




**Figure 2: Concentrations of key types of VOCs species (a) and their contribution to TVOCs, $L_{OH}$, OFP, SOAP and Toxicity (b).**

**Table 1: Comparison of OVOC concentrations in different sites**

| Site | Site Type | Period | Acetaldehyde | Acrolein | Propanal+Acetone | MEK | MACR+MVK | (MACR+MVK)/ISOP |
|------|-----------|--------|--------------|----------|------------------|-----|----------|------------------|
| Beijing [a] | Urban | 2018.5-2018.6 | 2.90 | - | 4.16 | 1.00 | 0.43 | 0.62 |
| Beijing [b] | Urban | 2020.6.29-7.30 | 3.42 | 0.29 | 4.07 | 0.65 | 0.46 | 1.84 |
| Xianghe [c] | Suburban | 2017.11-2018.1 | 1.22 | 0.13 | 1.20 | 0.32 | 0.06 | 1.50 |
| Lhasa* | Urban | 2021.5-2021.6 | 3.59 | 0.25 | 3.25 | 0.35 | 0.27 | 1.50 |
| Lulang** | Forest | 2021.4-2021.5 | 1.96 | 0.19 | 2.29 | 0.20 | 0.13 | 6.84 |
| Menyuan [d] | Background | 2013.9 | 0.94 | - | 1.66 | - | - | - |
| Arctic [e] | Background | 2018.4-2018.10 | - | - | 0.61 | 0.03 | - | - |

a. (Huang et al., 2020), b. (Zhang et al., 2022) c. (Yang et al., 2019), *. (This study), **our measurement in 2021, d. (Zhao et al., 2020), e. (Pernov et al., 2021)

The $L_{OH}$, OFP, SOAP, and Toxicity of VOCs were evaluated. As shown in Fig. 2 (b), alkenes and OVOCs accounted for over 90% of the $L_{OH}$ and over 80% of the OFP, suggesting their key role in perturbing the photochemistry of $O_3$. Acetaldehyde and acetone were the top two OVOC

species (Table S1), and their concentrations reached 3.59±3.44 ppb and 2.53±0.83 ppb, respectively, accounting for 19.2% and 13.5%, respectively, of the TVOCs. Due to the relatively high reactivity of aldehydes, C3-C6 acetaldehydes were also the most important species in terms of $L_{OH}$ and OFP. The concentrations of alkenes were lower than those of alkanes but contributed more to $L_{OH}$, OFP, SOAP and toxicity. For example, due to its high activity, isoprene ranked 2nd in terms of $L_{OH}$. Although halohydrocarbons and biomass burning tracers of acetonitrile also comprised a considerable fraction of the TVOCs (14%), their inert nature inhibits their perturbation effects on the $O_3$ photochemistry. While

aromatics further contributed to 13% to the OFP, it absolutely dominated the SOAP. Alkenes, mainly isoprene, also contributed to 9% to the SOAP. Therefore, alkenes, OVOCs and aromatics are the key VOCs perturbing the photochemistry. It was alkenes, OVOCs and aromatics that also contributed more than 78% to the toxicity. Specifically, the contributions of benzene, toluene, ethylbenzene, m,p-xylene, and o-xylene



(BTEX) and isoprene-derived OVOCs to both the SOAP and toxicity were high. Contribution of each important VOC species to TVOCs, $L_{OH}$, OFP, SOAP and Toxicity are shown in Fig. 3.

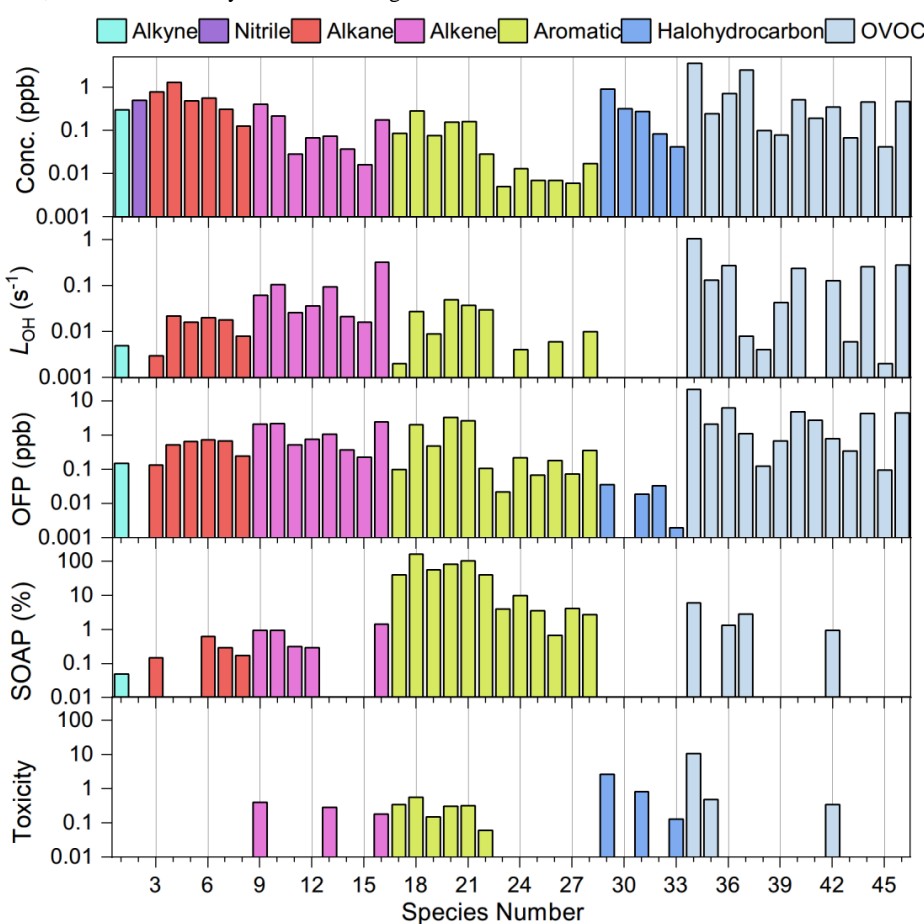


**Figure 3: 46 VOCs tracers selected for source appointment for PMF analysis and their contribution to TVOCs, $L_{OH}$, OFP, SOAP and Toxicity if available. At least one tracer for typical VOC emission sources and one VOC tracers from each type of VOCs was included. These selected VOCs contribute to over 90% of TVOCs, $L_{OH}$, OFP, SOAP and Toxicity. Species number is referred in Table S1.**




The concentrations of the TVOCs and major anthropogenic alkanes and aromatics measured in this study are approximately twice as low as those measured in the megacity of Beijing (Li et al., 2020), but several folds or even more than one order of magnitude higher than those measured at most other regional measurement sites across the TP (Xue et al., 2013; Xu et al., 2022) (Table 2), confirming the anthropogenic contribution to VOCs. The levels of alkenes, especially isoprene, are comparable to measurements in the megacity of Beijing (Li et al., 2020) or at a forested site

in Lulang (another measurement study within the @Tibet field campaigns), suggesting a possible biogenic contribution to VOCs, in addition to anthropogenic emissions. The OVOCs measured in this study were among the most abundant at all the urban, suburban, background and forested station sites listed in Table 1. As stated above, both the topography and high ultraviolet (UV) irradiation, accompanied with NOx emission, could encourage the accumulation of OVOCs in the Lhasa River valley. Compared to the three previous measurements of VOCs in Lhasa (Yu et al., 2001, 2022a; Guo et al., 2022), we measured more abundant OVOCs and therefore higher OVOC proportions. The relatively high abundance of

alkanes was captured by all VOC measurements in Lhasa. Specific OVOCs and aromatics were also comparable in concentration in all the measurements in Lhasa when these species were included (Table S2). Our online GC-MS deployment, with more comprehensive VOCs measurements, well represent the VOC pollution for Lhasa.

**Table 2 Comparison of VOCs concentrations in different sites**

| Site | Site Type | Period | Alkanes | | Alkenes | | | Alkyne | Aromatics | | | | |
|---|---|---|---|---|---|---|---|---|---|---|---|---|---|
| | | | Ethane | Propane | Ethene | Propene | Isoprene | Ethyne | Benzene | Toluene | Ethylbenzene | *m,p*-xylene | *o*-xylene |
| **Beijing [a]** | Urban | 2016.3-2017.01 | 6.00 | 4.40 | 4.00 | 1.00 | 0.30 | 2.80 | 1.00 | 1.20 | 0.30 | 0.70 | 0.30 |
| **Xianghe [b]** | Suburban | 2017.11-2018.1 | 6.02 | 2.98 | 4.05 | 0.92 | 0.04 | 2.13 | 0.92 | 0.97 | 0.36 | 1.02 | 0.32 |
| **WLG [c]** | Background | 2003.4-5, 7-8 | 1.50 | 0.28 | 0.13 | 0.03 | 0.01 | 0.39 | 0.09 | 0.18 | 0.02 | 0.12 | 0.05 |
| **Lhasa\*** | Urban | 2021.5-2021.6 | 0.78 | 1.28 | 0.40 | 0.22 | 0.18 | 0.30 | 0.09 | 0.28 | 0.08 | 0.16 | 0.16 |
| **Lulang\*\*** | Forest | 2021.4-2021.5 | 1.52 | 0.41 | 0.40 | 0.13 | 0.02 | 0.56 | 0.13 | 0.05 | 0.03 | 0.16 | 0.10 |
| **Nam Co [d]** | Background | 2020.8 | 0.62 | 0.20 | 0.21 | 0.10 | 0.07 | 0.10 | 0.54 | 0.41 | 0.02 | 0.02 | 0.02 |
| **Arctic [e]** | Background | 2008.8-9 | 0.62 | 0.09 | - | - | - | 0.06 | 0.02 | - | - | - | - |

a. (Li et al., 2020), b. (Yang et al., 2019), c. (Xue et al., 2013), *. (This study), **our measurement in 2021, d. (Xu et al., 2022), e. (Hellen et al., 2012)



### 3.2 Source apportionment of VOCs

Thirty-seven out of 98 VOC species were measured with a signal-to-noise ratio over 2, and another 31 species were measured with a signal-to-
noise ratio between 0.2–2. Apart from the signal-to-noise ratio, more considerations were applied for PMF VOC selection. The most abundant species of alkanes, alkenes, OVOCs and aromatics were selected, further weighted by their contribution to $L_{OH}$, OFP, SOAP and toxicity, and most importantly their tracing role to identify typical sources of VOCs (Fig. 3 and Table S1). Specifically, 13 most abundant OVOCs are selected, with acetaldehyde, acetone, propanal and n-butanal are the top four species. Among them, intermediate oxidation products of isoprene, MACR and MVK, are key tracers for biogenic sources (Guenther et al., 2012; Mo et al., 2018), and MTBE is a tracer of gasoline emissions (McCarthy et
al., 2013; Li et al., 2018). 12 most abundant aromatics are selected following OVOCs. Among them, benzene, toluene, ethylbenzene are tracers of biomass burning, vehicle emission, industrial emission and solvent usage (Liu et al., 2008; Yuan et al., 2010). A selected series of BTEX are source tracer of solvent usage (Yuan et al., 2010; Liu et al., 2020). 7 most abundance alkenes and 6 most abundant alkanes are further selected. Among them, ethane, propane and ethylene are important components of combustion emission plumes (Baudic et al., 2016). Isopentane and n-pentane are tracers of gasoline evaporation (Baudic et al., 2016; Liu et al., 2008). A series of tracers were also selected from Halohydrocarbons.
For example, 1,2-Dichloroethane and chloroform are associated with industrial solvents or additives (Cai et al., 2010). Ozone-depleting substances such as Freon 11 were recognized as background compounds, which were formerly used as refrigerants but are now largely banned (Saeaw and Thepanondh, 2015). Together with acetonitrile and acetylene, chloromethane is tracers of biomass burning (Liu et al., 2008; Chen et al., 2017). Finally, 46 out of 98 VOC species were selected to PMF analysis. These species contributed 94.8% to the TVOCs, 96.8% to $L_{OH}$, 96.9% to the OFP, 98.3% to the SOAP, and 97.5% to the toxicity. Inorganic species BC, $NO_x$, NO and CO were also included to help to discriminate the
transport sector from biomass burning and solvent usage, as distinct emission ratios of these inorganic species are well documented for these types of emission sources (Chen et al., 2017; Gentner et al., 2017). Finally, 50 chemical species were selected and included in PMF analysis and these 50 species could inform us of both the source identity and environmental significance of VOCs in the city of Lhasa (Fig. 3).





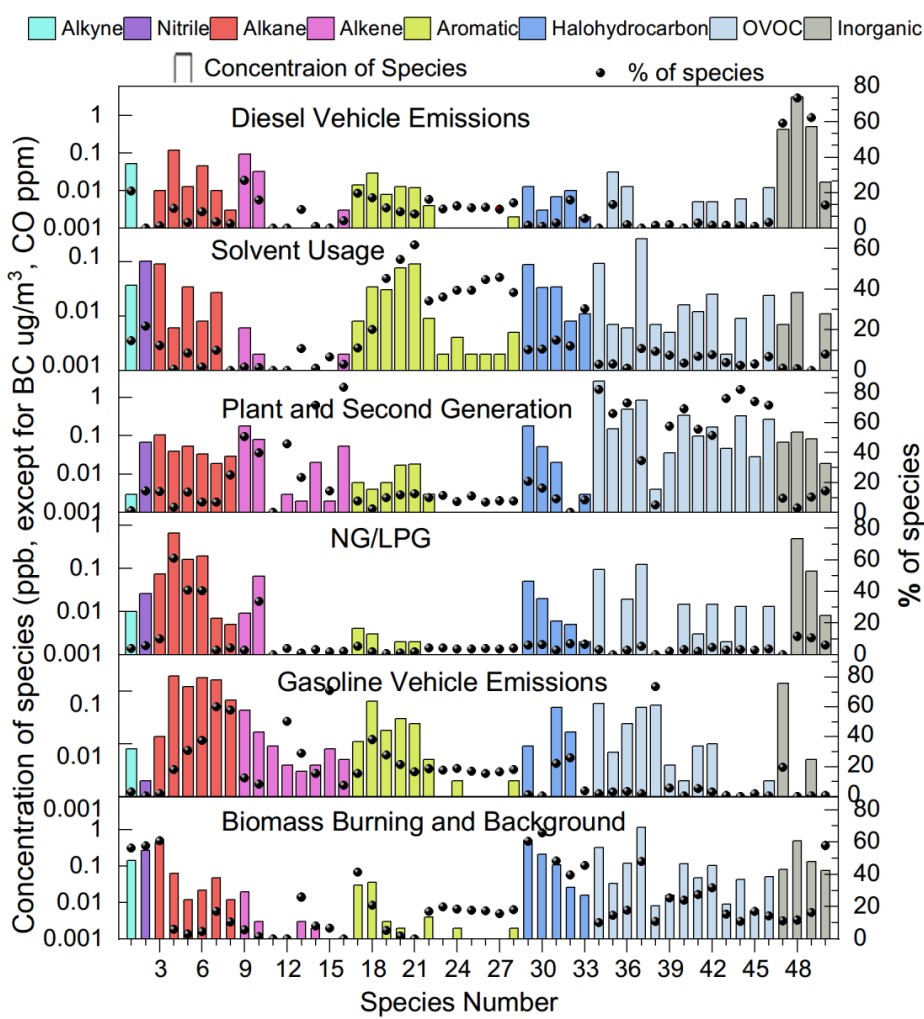

**Figure 4: Source factors by PMF decomposition and source identity attribution for the 4 inorganic tracers and 46 VOC tracers. (The concentration units for BC and CO were ug/m³ and ppm)**



PMF decomposed six factors (Fig. 4). The tracers of emission sources were normally well described by one to three of the factors. Factor one accounted for over 60% of BC and $NO_x$ but contributed much less to CO, leaving diesel vehicle emissions as the source identity (Gentner et al., 2017). A series of lines of evidence further support this appointment. A considerable contribution to ethylene, propene and some aromatics agrees

with the source spectrum of diesel vehicle emissions (Mo et al., 2016). The source spectrum of the decomposed diesel vehicle emission resembles the typical spectrum collected from on-road sampling (Fig. 5). As shown Fig. 6, ternary analysis of the benzene series also confirmed the contribution of diesel vehicle emissions to VOCs (Zhang et al., 2016). Finally, the diel profile of the decomposed diesel vehicle emissions reflected the early morning and later evening rush hours with heavy-duty trucks crossing the city, as the daytime prohibition policy of trucks has been implemented in Lhasa (Fig. 7). The overall contribution of diesel vehicle emissions to the TVOCs was minor, reminding us of the benefits

of the effective management of diesel vehicle emissions in Lhasa. Meanwhile, its contributions to aromatics, BC and $NO_x$ were considerable or even important. Although the daytime prohibition policy of trucks allocates diesel vehicle emissions in the early morning and the later afternoon, i.e., a pattern to minimize its perturbation effect on the photochemistry of $O_3$ and SOAs, the overall contribution of diesel vehicle emissions to the OFP, SOAP and toxicity underlines the need to further reduce these emissions. There is an ongoing discussion on the replacement of diesel vehicles by electric cars. While we still do not see the progress of this replacement in real life, electric trucks could surely reduce the emissions of

BC, NOx and aromatics and therefore should be encouraged in the highland city of Lhasa with strong solar radiation.





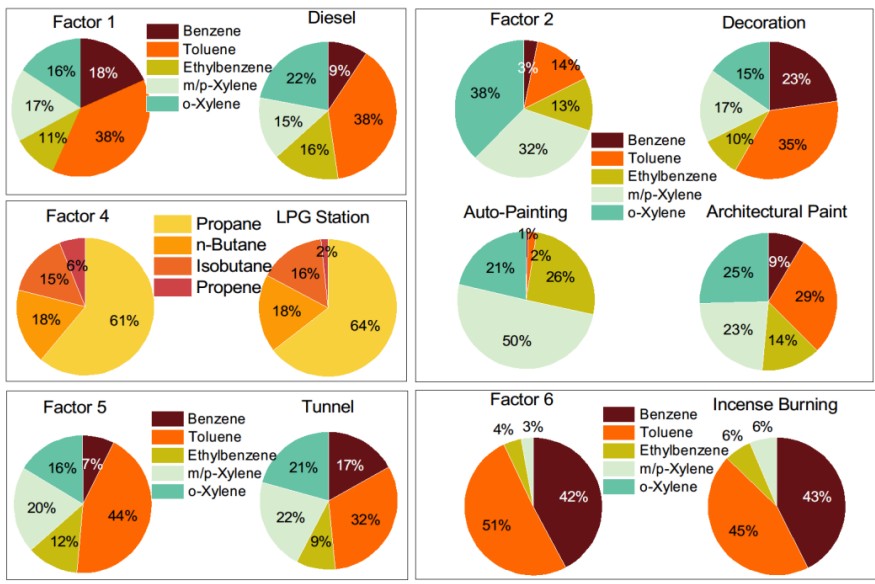

a

**Figure 5: Comparison of source spectrums decomposed by PMF with typical source spectrum measured in our study and reported in literature. The data of Auto-Painting and Architectural Paint were from (Yuan et al., 2010).**

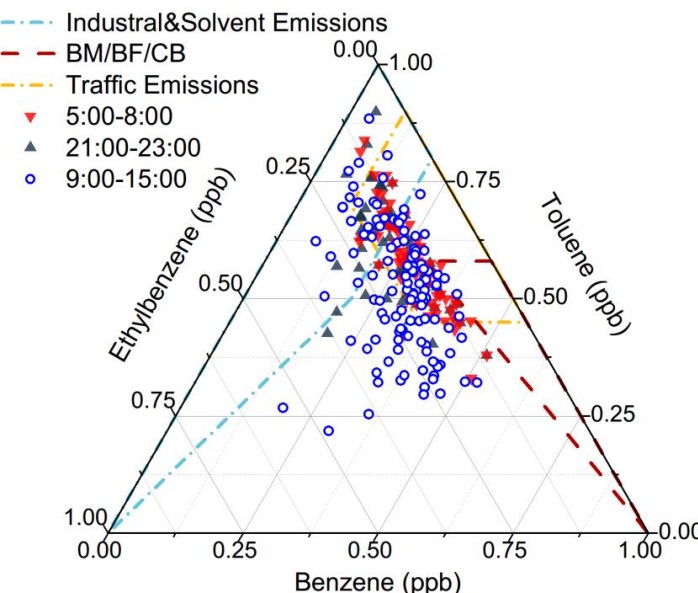


**Figure 6: Ternary diagram of benzene, toluene, and ethylbenzene. Plot suggests combined contribution to aromatics from Vehicle Emission, Solvent Usage and Biomass Burning. BM/BF/CB" indicates biomass/biofuel/coal burning. The wireframes include more than 90% of the scatter collected from refences (Zhang et al., 2016).**

Factor two featured a considerable contribution to aromatics and nearly no contribution to inorganic tracers (Fig. 4). The source spectrum was comparable to that of solvent usage emissions (Fig. 5) and consistent with the source profile of various solvent usages (Yuan et al., 2010; von Schneidemesser et al., 2016; Baudic et al., 2016). The ternary diagram of benzene, toluene, and ethylbenzene highlighted the contribution of solvent usage to aromatics. The diel profile of solvent usage emissions exhibited a nighttime peak (Fig. 7) as a result of the continuous emission and boundary layer regulation of VOC species. Although aromatics are not as reactive as alkenes or OVOCs, the associated SOAP, OFP and

toxicity currently indicate less effective management of solvent usage in Lhasa. Multiple painting activities in small businesses are observed in the city. The low pressure further increases the evaporation rate of solvent in this highland city. This requires strict implementation of regulation measures and work place evaporation confinement techniques to effective reduce the emission strength of aromatics and meanwhile to protect the health of workers.



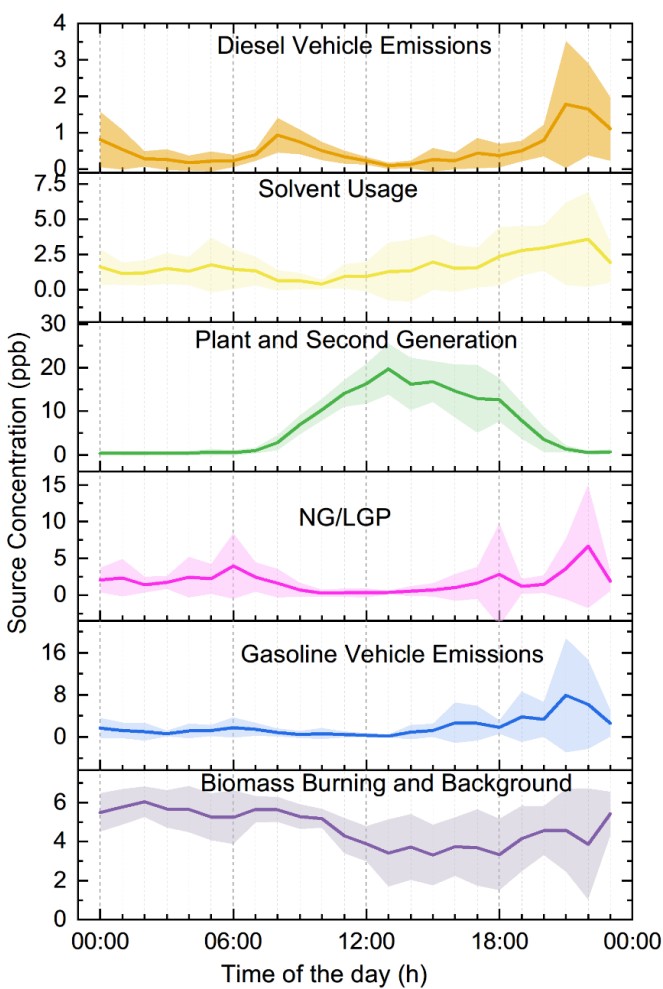

**Figure 7: Diel profiles of six decomposed factors.**

Factor three accounted for isoprene and most of the OVOCs. Acetaldehyde and acetone are photochemical intermediates of many VOC precursors (Mellouki et al., 2015). The other selected OVOCs, except for MTBE, retained a double-bound feature, indicating that alkenes were



their key precursors (Fig. 4) (Baudic et al., 2016). The typical daytime peak of OVOCs, as confirmed in Figs. 7-8, supported the source identity of biogenic emissions and secondary generation. Notably, several alkanes, alkenes and aromatics are not only tracers of anthropogenic sources but

also suspected precursors of these OVOCs. The contribution of anthropogenic sources to the factor of biogenic emissions and secondary generation could thus not be excluded. A detailed budget analysis might be helpful to further identify the source of these OVOCs.

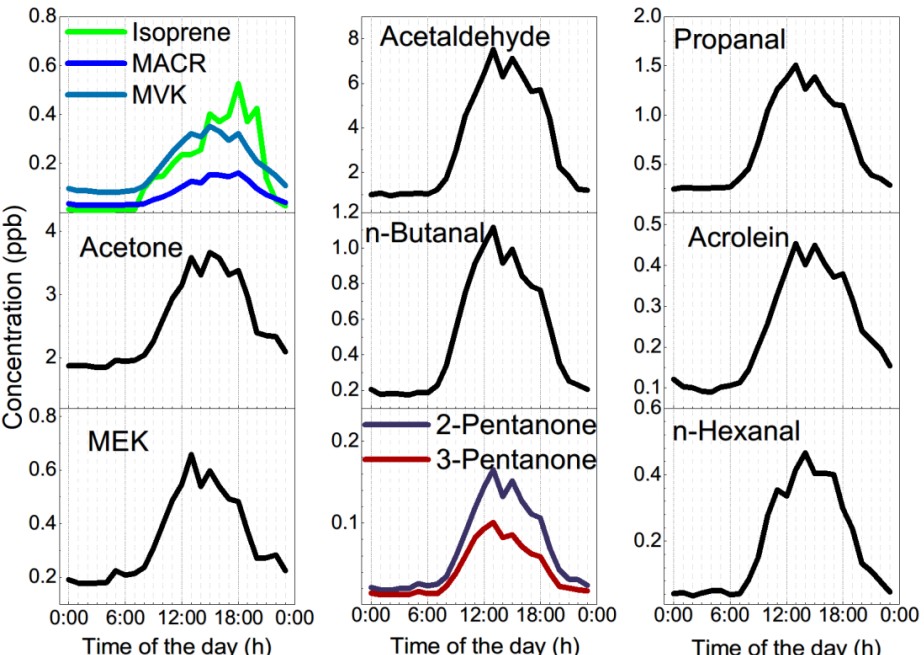

**Figure 8. Concentration diel profiles of key tracers of the Plant and Secondary Generation.**

Factor four featured a heavy contribution to propane, isobutene, n-butane and propene, which indicated the combustion of LPG for traffic and cooking (Fig. 4). A low contribution of LPG combustion to $NO_x$ also seemed reasonable, as central heating services cease at this time of the year (Liu et al., 2008; Lyu et al., 2016). The source spectrum of the decomposed LPG combustion resembled that sampled at the LPG station (Fig. 5). The diel profile of factor four also reflected the intensive usage of LPG from later night to early morning, likely for cooking purposes (Fig. 7). The low contribution of LPG combustion to most of these reactive VOCs verifies the effectiveness of the government policy to replace small



coal-fired boilers with LPG-fuelled central heating systems in the city. Notably, the considerable contribution of the LPG factor to NO$_x$ underlines the side effect of LPG-fuelled central heating.

Factor five featured a heavy contribution to a series of alkanes and a considerable contribution to several alkenes, MTBE and nearly all aromatics (Fig. 4). Gasoline vehicle emissions were therefore suspected (Liu et al., 2008; Lyu et al., 2016). Daytime tunnel sampling of the source spectrum confirmed this appointment (Fig. 5). The diel profile of gasoline vehicle emissions reflected the afternoon pattern of leisure life activities and the

long rush hour in Lhasa (Fig. 7). The negligible contribution to NO$_x$ was unexpected, which could be attributed to the uncertainty in this analysis. Further measurement of the vehicle emission ratio of NO$_x$ at this high altitude is encouraged. The vehicle population has increased sharply in Lhasa. As a recognized issue in air quality control, gasoline vehicle emissions or related fuel evaporation need better management. In contrast to electric trucks, electric cars have been a practical option to reduce the emission of both BC and all aromatics.

Factor six represented the last important source, which contributed evenly to inorganic tracers, alkanes, alkenes, aromatics and OVOCs. While

this indicates a mixture of varied source contributions, the heavy contributions to acetylene and acetonitrile highlighted biomass burning emissions (Fig. 4) (Akagi et al., 2011; Pernov et al., 2021). The diel profile suggested boundary layer regulation and further indicated a background feature of factor six. In fact, a series of regional sources could not be decomposed in our PMF analysis. Biomass burning has been a regional source of both VOCs and BC (Li et al., 2017). Our interview with local residents suggests the abandonment of biomass in household heating. However, biomass burning in the suburban area was observed. In addition, incense burning in Lhasa is another source of VOCs and BC

(Cui et al., 2018; Lu et al., 2020). Comparison of the decomposed source spectrum to our measurements of the incense burning emission spectrum indicates the potential contribution of this source to factor six. We therefore identified factor six as biomass burning and a background source. This source contributed considerably to BC, NO$_x$ and aromatics and therefore must not be negligible.

A previous source appointment by Guo et al. (2022) confirmed the contribution of diesel vehicle emissions, solvent usage, natural gas (NG)/LPG, gasoline vehicle emissions and biomass burning and background sources. However, due to the failure in OVOC measurements therein, secondary

generation was not decomposed. This led to underestimation of the TVOCs, L$_{OH}$, OFP, SOAP and toxicity and negligence of biogenic emissions. A more comprehensive understanding of the key species and key source of VOC should be updated according to our analysis.

### 4. Implications

Our measurement and source appointment based on online GC–MS of VOCs clearly showed the contribution of six typical sources to their

concentration, environmental impact and health effect. Figure 9 further shows the diel profiles of the contributions of the six decomposed VOC sources. TVOC shows bi-peak patterns at noontime and in the later afternoon. The former was dominated by OVOCs, and the latter was dominated by the combination of LPG, solvent usage, and gasoline vehicle emissions. Due to the chemical reactivity of OVOCs and alkenes,



biogenic emissions and secondary generation sources dominated $L_{OH}$, OFP and toxicity and contributed to the SOAP (Fig. 10). In regard to $O_3$ production, not only the absolute contribution but also the diel profiles of the sources are important. The overlap of the abundant OVOCs and

alkenes with the photochemical production of $O_3$ during the daytime further amplifies the role of biogenic emissions and secondary generation sources in $O_3$ pollution in Lhasa. Nighttime and early morning hour accumulation of other OVOC precursors, such as alkanes and aromatics, could also contribute to the daytime accumulation of OVOC and $O_3$ production via photochemical oxidation. However, the role of these photochemical oxidation reactions could not be determined in PMF analysis. A full $O_3$ budget analysis with a chemical model is planned to better quantify the role of varied VOC sources in $O_3$ production. Aromatics were the major contributors to the SOAP. Source appointment suggested the

combined contribution of solvent usage, diesel vehicle emissions and biomass burning and background sources. While boundary layer development during the daytime could dilute aromatics and other VOCs, the SOAP of these three sources could not be fully determined. Deduced based on the accumulation of these abundant OVOCs, the accumulation of the oxidative intermediates of aromatics was also expected but was not measured by our GC–MS instrument. The suppression of aromatics contribution into the SOAP due to boundary layer dilution could potentially be offset by the accumulation of the oxidative intermediates of aromatics. This assumption will also be further examined via MS aerosol chemical

component analysis and chemical model examination. In contrast, OVOC contributions to the SOAP were confirmed, although photochemical intermediates of aromatics, at least these primary intermediates, were not measured by GC–MS.



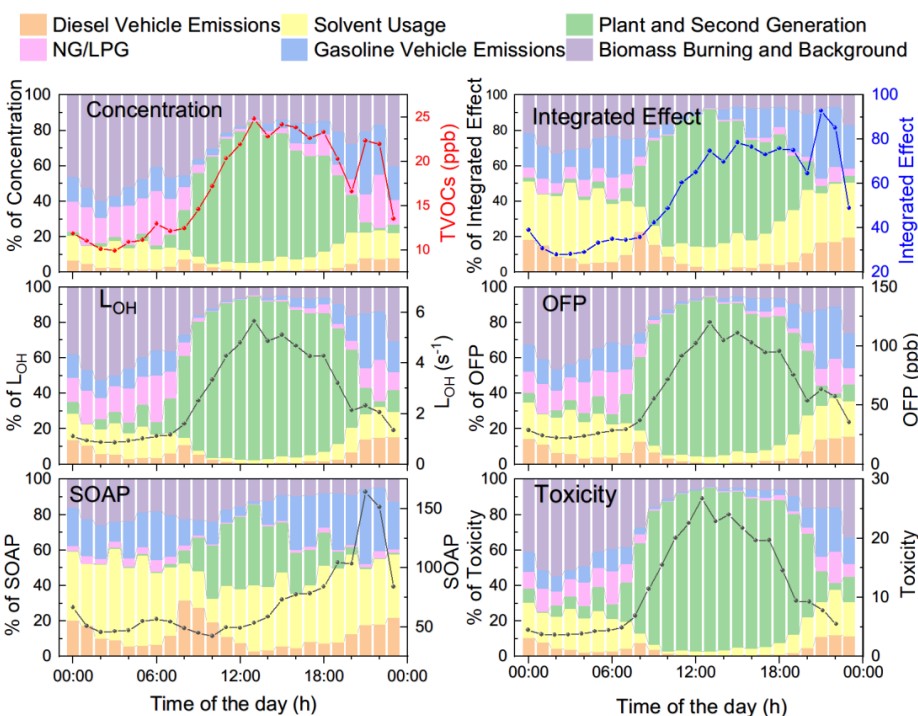

400 **Figure 9: Diel profiles of key sources of VOCs and their contributions to TVOCs, $L_{OH}$, SOAP, Toxicity, OFP, and Integrated Effect.**



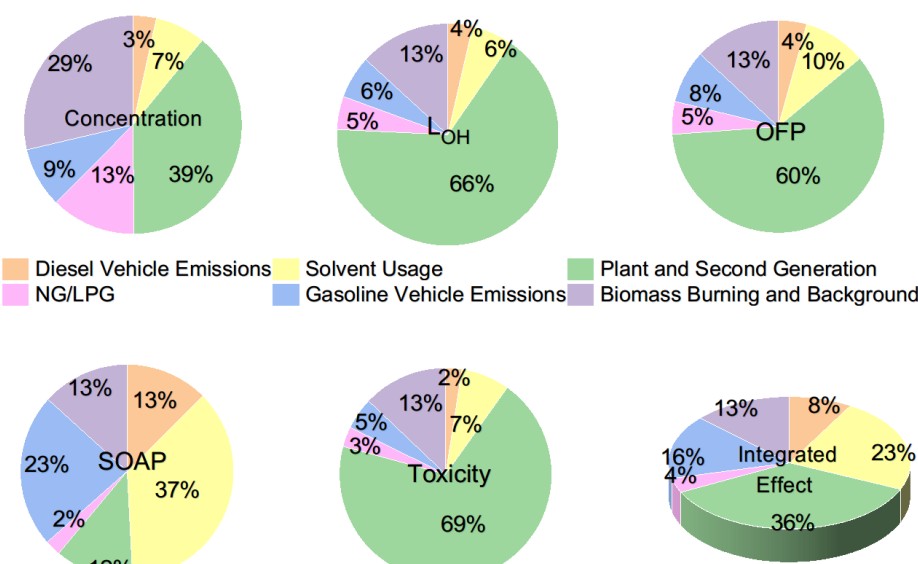

**Figure 10: Average contributions of key sources to TVOCs, $L_{OH}$, SOAP, Toxicity, OFP, and Integrated Effect.**

The role of OVOCs highlights that photochemical oxidation of primarily emitted VOCs in the highland city of Lhasa further amplifies the role of primary sources in $O_3$ photochemical production. Especially in the highland city of Lhasa located in the Lhasa River valley, both active photochemistry powered by high UV and high $NO_x$ levels and the topography are favourable conditions for the accumulation of OVOCs but unfavourable conditions for $O_3$ pollution control. There has been an ongoing discussion on how the natural cycle of VOCs is amplified by the anthropogenic emission of $NO_x$ in terms of the SOA yield (Ziemann and Atkinson, 2012; Priestley et al., 2021; Luo et al., 2021; Zhan et al.,

2021). Our data show that not only the SOAP but also $L_{OH}$, OFP and toxicity are amplified by the mutual feedback among the biogenic and anthropogenic emissions of VOCs and anthropogenic emissions of $NO_x$. As such, simultaneous emission reduction in VOCs and $NO_x$ should be considered. Transport sector emissions and biomass burning and background emissions should be targeted for their contributions to VOCs, $NO_x$ and BC. Due to the potential contribution of biogenic alkenes to OVOCs, the benefits of biogenic emission reduction could be high. However, Lhasa city is devoted to planting trees and managing dust pollution along the Lhasa River Valley. Side effects on biogenic emission and $O_3$

pollution control should be considered.



**Author contribution**

C.Y. and W.L. designed the research. S.G., J.W. C.Z., and Y.C. carried out the field measurements. S.G. performed data analysis and interpreted the data. C.Y., S.G., W.L., and J.W. prepared the manuscript with contributions from all co-authors.

**Competing interests**

The authors declare that they have no conflict of interest.

**Acknowledgement**

This work was supported by the National Natural Science Foundation of China (Grants No. 41875151 and 21876214), and the Second Tibetan Plateau Scientific Expedition and Research Program (Grants No. 2019QZKK0604).

**Special issue statement**

**Data Availability Statement**

All data used in this paper are collected during the @Tibet campaigns in 2021 and are publicly available. The merged data set is managed by authors and the TP data center and could be downloaded currently at https://data.tpdc.ac.cn/zh-hans/disallow/49f352a2-a160-49aa-a955-f5a0bb397c93.

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
