# Peer review of "Measurement Report: Source apportionment and environmental impacts of VOCs in Lhasa, a highland city in China"

_EGUsphere, 2022_

## Author Comment (AC1)

**Response to referees' comments**

**Response to comments by referee RC1**

**General Comments:**

The submitted manuscript by Ye et al concerns how VOC measurements in Lhasa (a city in China within the Tibetan Plateau) impact ozone and secondary aerosol production. It also reported the calculated toxicity impacts of the VOC compounds measured. To find the source apportionment of VOCs, the EPA's PMF model was employed. The manuscript does a good job of explaining the need for reporting VOCs in Lhasa due to its unique elevation and geography, and high populations. The field's knowledge of VOC sources and emissions for each compound is well-researched and cited.

Response: We appreciate your constructive comments. Point-to-point response to your comments are listed below. The corresponding changes were highlighted in our revised manuscript.

**Specific comments:**

I would include more information about the meteorology conditions during the campaign. What were the average daytime and nighttime temperatures? Chilly nighttime temperatures would provide more evidence of the NG/LGP factor peaking at the 17:00 and 22:00 hours as people used heat to keep their homes warm.

Response: As can be seen from Fig. 1, the average temperatures were ～20 ℃ during the day and ～15 ℃ at night during the observation period. Central heating had been ceased. Public transportation (buses and taxis) and cooking appeared to be major sectors consuming NG/LPG.

I would appreciate more context and background of the integrated vs relative toxic effect in the introduction or methodology. What metrics and analysis goes into the "Toxicity Grade". Also, while you point out that VOCs in Lhasa have been understudied and so much is unknown, I would discuss how Ozone and SOA formation in other (and better characterized) cities would compare with Lhasa given its climate and topography.

Response: We revised the text in lines 198-210 in the revised manuscript as follow.
The relative toxicity effect and integrated effect assessment were performed referencing the methods developed by Niu et al. (2016). In their method, the multi-effects of OFP, SOAP and toxicity had been considered together to evaluate an integrated effect assessment of VOCs.
Relative Toxicity Effect = [VOCs]×Toxicity Grade                    (4)

Where, toxicity grades for VOCs species adopt the definition by Niu et al. (2016), in which four toxicity grades are classified on the basis of information on the carcinogenic, teratogenic, mutagenic properties of different VOC species originating from both the European Commission and the International Agency for Research on Cancer (IARC). Toxicity grade 1 represents IARC group 3; grade 2 allocates to IARC group 2B which is possibly carcinogenic to humans; grade 3 represent IARC group 2A, probably carcinogenic to humans; grade 4 represent group 1 carcinogens, mutagens, teratogens, or highly toxic to humans. The toxicity grades and concentrations were multiplied to estimate the relative toxic effect of VOCs.

According to Niu et al (2016), the integrated effect was calculated by eq. 5. The weightings of these adverse effects of OFP, SOAP and toxicity were assigned by expert scoring. OFP and SOAP are crucial for the formation of $O_3$ and $PM_{2.5}$, assigned 40% weight separately. The weighting of VOCs relative toxicity effect was assigned as 20% in view of the error.

$$Integrated\ Effect = 0.4 \times Relative\ OFP\ contribution + 0.4 \times Relative\ SOAP\ contribution + 0.2 \times Relative\ Toxicity\ Effect\ contribution \quad (5)$$

Also, to compare ozone photochemistry between Lhase and Beijing is a terrific idea. We calculated ozone production by constructing a observation-prescribed MCM model. However, a comprehensive discussion on the the ozone production photochemistry in Lhase appeared to be too aggressive or a bit crowd to be held in this manuscript. Alternatively, we stated "A full $O_3$ budget analysis with a chemical model is planned to better quantify the role of varied VOC sources and the accumulation of OVOCs in $O_3$ production." Taking into our data available for now, SOA chemistry is a much more complex issue which surely needs more research attention. In addition to the impression on dominating role of primary aerosols, such as dust and BC, we did oberve high abundance of SOA components as a prelimnary impression from our EESI-LToF-MS data. We will follow the referee's advice to further look into the detailed SOA photochemistry.

While the PMF model was used appropriately, further discussion is needed with some of the interpretations and conclusions. In Fig. 5 (the source spectrum pie chart plot), I had issue with just using the few aromatic compounds for most of the comparison between your PMF results and literature source spectrums. Since all the aromatic compounds you included (xylenes, ethylbenzene, toluene, benzene) have similar sources (incomplete combustion, solvent use), the proportion of just those 5 compounds is suspect for declaring a confident source profile. Instead, I suggest including a wider range of compounds in your source spectrum plot, with a wider range of compound sources (volatile chemical products, biogenic compounds, solely solvent compounds). For example, I would show the source spectra and results of non-VOC pollutants (CO, NOx BC, NOx, NO) and the tracer compounds (1,2-Dichloroethane and chloroform, Isopentane and n-pentane) instead of just aromatics.

Response: We added more species and sources in Fig. 5 in the revised manuscript. The new Fig. 5 in the revised manuscript shown below shows the PMF-decomposed source spectrum, i.e., the fraction of major tracers, are comparable with typical ones. This point is also the main message passed by the original Fig. 5. However, we did see very different source spectrum of biogenic

VOCs from MEGAN's suggestion. The major reason is that we did not quantify the terpene species nor measure the plant source spectrum due to lack of corresponding calibration standards for our GC-MS. In addition, biogenic emission in alpine areas like Lhase needs specific research validate since MEGAN emission inventory shows some bias for another apline site in the the tibetan plateau (new data, not shown here).

[Figure]

**Figure 5. Comparison of source spectrums decomposed by PMF with typical source spectrum measured in our study and reported in literature. The data of Auto-Painting and Architectural Paint were from (Yuan et al., 2010). The biogenic data is from global data set of biogenic VOC emissions calculated by the MEGAN model (Sindelarova et al., 2014).**

Given more and more research showing the impact of volatile chemical product and human emissions (such as personal care products) in urban areas, did you find any contribution to that in your study?

Response: Yes, it has been reported that volatile chemicals (VCPs) replace transportation sources as the largest petrochemical source in densely populated areas in the United States (McDonald et al; 2018; Gkatzelis et al., 2021; Van Rooy et al., 2021). As for Lhase, VCPs might not be as important as in the literature, based on our observation on personal care product usage in local residents.

Representative species such as silanes and ethanol in personal care products were not qualitatively quantified in our methodology, so it is not possible for us to quantify the contribution of VCP to atmospheric VOCs. It would surely be interesting to include more measurements of VCPs given another chance to conduct field campaign in Lhasa in the futher.

Gkatzelis, G., Coggon, M. M., McDonald, B. C., Peischl, J., Aikin, K. C., Gilman, J. B., et al. (2021). Identifying Volatile Chemical Product Tracer Compounds in US Cities. Environmental Science & Technology, 55(1), 188–199. https://doi.org/10.1021/acs.est.0c05467

McDonald, B. C., de Gouw, J. A., Gilman, J. B., Jathar, S. H., Akherati, A., Cappa, C. D., et al. (2018). Volatile chemical products emerging as largest petrochemical source of urban organic emissions. Science, 359(6377), 760–764. https://doi.org/10.1126/science.aaq0524

Van Rooy, P., Tasnia, A., Barletta, B., Buenconsejo, R., Crounse, J. D., Kenseth, C. M., et al. (2021). Observations of Volatile Organic Compounds in the Los Angeles Basin during COVID-19. Acs Earth and Space Chemistry, 5(11), 3045–3055. https://doi.org/10.1021/acsearthspacechem.1c00248

For your "Plant and Second Generation" source factor, I would suggest renaming it "Sunlight-impacted" or something similar. Or at the very least, discuss how isoprene is very sunlight-dependent (very correlated with photosynthetically active radiation). Ozone formation is dependent on UV sunlight, so it makes sense that isoprene and ozone-related compounds are correlated. Further, did you measure any other known biocgenic compounds (alpha/beta pinene)? Comparing their PMF results with the isoprene results and the resulting "Plant and Second Generation" would be helpful in separating further sources. The "Plant and Second Generation" matches past reports of diel trend of ozone and isoprene.

Response: We accepted Sunlight-impacted. In Figure 8, we added the diurnal variations in the short-wave radiation and ozone. Isoprene and especially its intermediate product MVK, MACR closely correlate with solar radiation intensity and ozone. In the following chemical model study on ozone photochemisty, we will further put our emphasis on the role of the emission diel pattern, in addition to the emission strength, in ozone production. Our method cannot quantify terpene species, so information on emissions from other biological sources is not available.

For the Gasoline Vehicle Emissions, I would appreciate more explanation of the diel profile. In lines 358-360, you mention the "afternoon leisure activities and evening rush hour", but there is no spike in the diel trend in the morning time during the morning rush hour when a similar amount of traffic would be on the road compared to the evening rush hour. To my eye, the Diesel Vehicle emissions spike at 8:00-9:00 and at 20:00 looks a lot like morning and rush hour emissions.

Response: The morning rush-hour spike of gasoline vehicle emissions is less evident, which might be related to both the relatively-relax work schedule and much high boundary layer in the morning than that at night. The radiation and temperture there is more favorable for evening activities, peaked at 20:00, when the mixing layer height was also much lower than that at morning (see Figure 4e in Guo et al., 2022). The city was also influenced by heavy-duty trucks. The trucks follow the daytime prohibition policy. The truck driver likes city supply, but have to leave the city

in the early morning, resulting in a more evidental morning rush hour spike of diesel vehicle emissions.

In Section 3.3, you discuss the role of incense burning as a contribution, but the papers you cite report incense enhancements during religious holidays and for mosquito repellant. Were there religious holidays during your analysis (and did you see subsequent increases as Cui et al 2018 did)? Also, in line 369 you mentioned how "Biomass burning in the suburban area observed". Does that mean visually (as in, you saw smoke)? The ternary diagram (Fig 6) shows very few data points in the biomass/biofuel/coal burning regime. Please include some text in the manuscript describing the comparison between your obervations and the results.

Response: During the observation period, there were no religious festivals. Temples and other religious sites were crowded with tourists in summer and incense burning activities occurred everywhere and anytime in the city. In suburban and rural areas on the outskirts of Lhasa, biomass burning activities, such as wood and cow dung burning, is very common for cooking or heating even for now, which contributed a regional background of Lhasa pollution. In Figure 6, the points of benzene, toluene and ethylbenzene distributed in the intersection area crossing the traffic, biomass combustion and solvent emission areas, indicating a mixed influence of the three types of emissions.

**Technical Corrections**

1. Lines 160-164 specify the time resolution of the NOx, BC, O3, CO instrumentation.

Response: We added the time resolution in the revised paper.

2. Lines 110 provide more detail about in which direction the O3 precursors might dramatically shift from past levels (while the precursors become less abundant vs. more abundant)?

Response: According to the MEIC inventory data, it showed that VOCs emissions in Tibet went through a process of decreasing from 2008 to 2013 and then increasing to 2017 (Li et al., 2019).

3. Lines 115-124: Methodology of past GC-MS measurements is not necessary for introduction. Instead, incorporate into your methodology section (Section 2.2)

Response: Here we just introduce the previous measurements with different methods. The methods are not the measurement methods applied in the paper.

4. Line 202: You use S/N without defining signal to noise (although you do use it later in the paper)

Response: We added the definition of S/N in the revised paper.

5. Line 226: Add "concentration" after CO

Response: accepted.

6.  Table 1: Add units

Response: accepted.

7.  Line 259 Name the isoprene-derived OVOCs that were **as toxic as BTEX** (that finding is quite significant, in my opinion)

Response: We double-checked and delete the phrase. As a matter of fact, we assumed multiple precursors for OVOCs, as isoprene could merely account for a minor fraction of OVOCs in our chemical model (not shown).

8.  Line 267: Say actual value of increase (2x, 12x, etc) versus "several folds or even more than one order of magnitude"

Response: We added after this sentence: For example, the propane, propene, and isoprene concentrations in Lhasa were 4, 7, and 18 times higher than that in WLG, and were 6, 2, and 2.5 times higher than that in Nam Co, respectively. M, p-xylene and o-xylene were 8 times higher in Lhasa than in Nam Co.

9.  Line 274: Specify what the "relatively high abundance in Lhasa" is relative to

Response: The relatively high abundance of alkanes compared to other VOC compounds except OVOC was captured by all VOC measurements in Lhasa.

10.  Table 2: Add units

Response: added in the revised paper.

11.  Line 290-291: "Among them, benzene, toluene, ethylbenzene are tracers of biomass burning, vehicle emission, industrial emission and solvent usage (Liu et al., 2008; Yuan et al., 2010). A selected series of BTEX are source tracer of solvent usage (Yuan et al., 2010; Liu et al., 2020)" These sentences are redundant.

Response: The two sentences have been combined.

12.  Line 342: double bond feature, not double bound

Response: corrected in the revised paper.

13.  Line 368: Were the interviews with residents scientifically significant? Anecdotal? I would suggest not including that or citing a better source for the reduction in biomass in household cleaning.

Response: We added data form local government and revise the sentence. By the end of 2014, Lhasa City basically achieved nearly full (98%) coverage of heating by natural gas (from

government data), suggesting a negligible biomass burning for household heating, which also confirmed by our interview with local residents.

---

## Author Comment (AC2)

**Response to referees' comments**

**Response to comments by referee RC2**

**General comments**

The manuscript "Measurement Report: Source apportionment and environmental impacts of VOCs in Lhasa, a highland city in China" by Ye et al., presents VOC and other species measured in the highland city of Lhasa, China and identifies the major emission sources through PMF analysis.

The paper is very well written and organized. The information and discussion provided is very valuable and backed up by sounded experimental data. In my opinion the scientific community will greatly benefit from the publication of this study.

The manuscript only needs a minor revision before being ready.

Response: We appreciate your constructive comments. We used them to improve our

manuscript.

I think it would be useful to have more background and context about the parameters used in equations 3, 4, and 5 and some additional discussion on the importance of such assessments (especially for the toxicity parameters).

Response: accepted. We added more information in the revised manuscript.

The authors mention the collection of canisters in addition to the online GC system. It seems that the bulk of the discussion was based on the online result and it's not clear how the grab samples were utilized.

Response: Negative-pressure instantaneous sampling of whole-air samples was performed at Lhasa in different locations, including tunnels, diesel vehicle emission-impacted roadsides, refueling stations, renovation sites, and incense burning locations during the observation period. Totally 36 canisters were collected. The samples were analyzed via GC–MS within 10 days. Here, we compare the measured source spectrum with the source apportionment results of PMF analysis (see Fig. 5).

**Minor Comments**

1. A long list of acronyms is introduced **in the abstract** without being spelled out first: $O_3$, $NO_x$, OH, PMF, NG/LPG, BC, TP.

Response: We checked and corrected in the revised paper.

2. Line 62: change LOH to $L_{OH}$

Response: accepted.

3. Line 72. Is the Okamoto and Tanimoto study specific for Chinese mountain sites?

Response: Okamoto and Tanimoto study included Mt. Waliguan in the Tibet Plateau.

4. Sentence starting line 74: "Among these studies …" needs references

Response: we added references of Liu et al., 2013; Ran et al., 2015; Cui et al., 2018.

5. Line 88. Indicate population of Lhasa

Response: added. In the 2020 census, Lhasa has a permanent population of about 870,000, with an increase of 55% compared to 2010.

6. Line 90. Please explain in different ways what you mean by "The city is not green or hot"

Response: Large areas of high mountains around Lhasa are bare, and vegetation coverage is less than 10% for sure. Coupled with low temperature throughout the year, plants are not flourishing, hence high biogenic emissions are therefore not notably considered.

7. Line 93 and further down in the introduction: is there any data on how elevated O3 is in the region? The authors mention multiple times "aggravated ozone pollution" without giving an idea on what the concentrations are for the reader to evaluate.

Response: Compared to the levels three decades ago, enhanced photochemical production of $O_3$, for example, with an average increase of 10 ppb $O_3$ in 2012 when comparing with the afternoon $O_3$ peak in 1998, and therefore aggravated $O_3$ pollution were recorded in 2012 (Ran et al., 2014). Aggravated $O_3$ pollution even lasts to the present day, with a slower increasing slope over the last few years (Yin et al., 2019). From 2005 to 2018, Lhasa had an average of 19 days with the maximum daily 8-h O3 exceeding 100 $\mu g/m^3$ per year (Li et a, 2020).

Li, R., Zhao, Y., Zhou, W., Meng, Y., Zhang, Z., Fu, H.: Developing a novel hybrid model for the estimation of surface 8 h ozone (O3) across the remote Tibetan Plateau during 2005–2018. Atmos. Chem. Phys., 20, 6159-6175. https://doi.org/ 10.5194/acp-20-6159-2020.

8. Canister Sampling – line 165. When were the cans collected? How many? Where? How long was the sampling time?

Response: Negative-pressure instantaneous sampling of whole-air samples was performed at Lhasa in different locations for typical emissions, including tunnels, diesel vehicle emission-impacted roadsides, refueling stations, renovation houses, and incense burning locations during the observation period. Totally 36 canisters were collected with three sampes for each types of emission sources. The samples were analyzed via GC–MS within 10 days.

---

## Author Response (AR2)

**Response to editor' comments**

**Comments:**

Thank-you for submitting your detailed responses to the comments made by the reviewers. I am happy to recommend publication in ACP following a couple of minor revisions:

Line 418 - could you provide a citable reference to the government report here?

With regards to the specific comment by RC1 on how ozone and SOA formation in other cities would compare with Lhasa given its climate and topography, I appreciate that it is beyond the scope of this paper to include box model results investigating the ozone budget. I think it would be useful though, in section 4, to draw the reader's attention once more to the comparisons already made in terms of the VOC distributions in Lhasa and other locations (shown in Table 1 and 2).

Response: Thank you. Following your suggestions, we revised our manuscript as below:

In line 118, we added the citable reference "Ministry of Ecology And Environment of the People's Republic of China, Report on the State of the Environment in China 2015,http://english.mee.gov.cn/Resources/Reports/soe/Report/201706/P02017061450 4782926467.pdf, latest access on 2023-6-12"

In line 381, we added the statement "Our results confirmed the anthropogenic contribution to VOCs in Lhasa. Both the topography and high ultraviolet (UV) irradiation, accompanied with anthropogenic $NO_x$ emission, also encouraged the accumulation of OVOCs in the Lhasa River valley. As a matter of fact, OVOCs accounted for the largest proportion of the TVOCs and the total OH reactivity. The much higher abundance and stronger contribution to the total OH reactivity of OVOCs than that of their precursors also characterized the photochemical decaying features of for example alkenes and long-chain alkanes in the Lhasa River valle. Effective photochemical production and accumulations of these measured OVOCs would perturb the budget of radicals and promote the photochemical production of $O_3$ and SOA, in addition to these primary emission of those VOC precursors."